# Insights into the Clinical, Biological and Therapeutic Impact of Copy Number Alteration in Cancer

**DOI:** 10.3390/ijms25136815

**Published:** 2024-06-21

**Authors:** Shannon L. Carey-Smith, Rishi S. Kotecha, Laurence C. Cheung, Sébastien Malinge

**Affiliations:** 1Telethon Kids Cancer Centre, Telethon Kids Institute, Perth, WA 6009, Australia; shannon.carey-smith@telethonkids.org.au (S.L.C.-S.); rishi.kotecha@health.wa.gov.au (R.S.K.); laurence.cheung@telethonkids.org.au (L.C.C.); 2Curtin Medical School, Curtin University, Perth, WA 6102, Australia; 3Department of Clinical Haematology, Oncology, Blood and Marrow Transplantation, Perth Children’s Hospital, Perth, WA 6009, Australia; 4UWA Medical School, University of Western Australia, Perth, WA 6009, Australia; 5Curtin Health Innovation Research Institute, Curtin University, Perth, WA 6102, Australia

**Keywords:** genetic, cancer, prognostic, targeted therapy

## Abstract

Copy number alterations (CNAs), resulting from the gain or loss of genetic material from as little as 50 base pairs or as big as entire chromosome(s), have been associated with many congenital diseases, de novo syndromes and cancer. It is established that CNAs disturb the dosage of genomic regions including enhancers/promoters, long non-coding RNA and gene(s) among others, ultimately leading to an altered balance of key cellular functions. In cancer, CNAs have been associated with almost all steps of the disease: predisposition, initiation, development, maintenance, response to treatment, resistance, and relapse. Therefore, understanding how specific CNAs contribute to tumourigenesis may provide prognostic insight and ultimately lead to the development of new therapeutic approaches to improve patient outcomes. In this review, we provide a snapshot of what is currently known about CNAs and cancer, incorporating topics regarding their detection, clinical impact, origin, and nature, and discuss the integration of innovative genetic engineering strategies, to highlight the potential for targeting CNAs using novel, dosage-sensitive and less toxic therapies for CNA-driven cancer.

## 1. Introduction

Copy number alterations (CNAs) are a hallmark of cancer, with approximately 25% of the genome of all tumour cells containing either chromosomal amplifications or deletions somatically acquired during cancer development [1,2,3]. However, while these alterations are clonally selected during tumourigenesis, their role, clinical impact, associations with specific cancers and specific tumour subtypes, and whether they are therapeutically targetable, remains elusive. Further emphasising their influence on cancer development, some congenital and de novo syndromes characterised by specific germline alterations, termed copy number variation (CNV), have a different incidence of cancer compared to the general population, indicating that these gains or losses of genetic material can either protect or promote tumourigenesis. The other chromosomal alterations commonly seen in cancer are inversions and translocations; however, in most cases, they do not modify the gene copy number. 

Mechanistically, CNA/CNV arises from a myriad of mechanisms that result in small to large genomic deletions or amplifications, and by definition lead to an altered dosage of genetic material such as regulatory elements (enhancers/promoters, long non-coding RNA, microRNA among others), non-protein-coding and protein-coding genes. Depending on the size (from 50 base pairs (bp) to an entire chromosome), the genomic location altered (rich or poor in regulatory elements or genes) and the magnitude (number of copy loss or gain of a genomic region), this may affect the expression of several tumour suppressor and tumour promotor genes that will ultimately impact cellular fitness, cell proliferation, differentiation, maturation, survival and response to treatment [4,5,6,7]. This becomes even more complex as, in most cases, tumour cells have more than one CNA, and thus together can lead to a precise gene expression profile that has clinical implications [8,9]. Notably, there is also a high level of intra-tumoural heterogeneity in most cancers [10], and CNAs alone may not be sufficient to drive tumourigenesis but rather cooperate with other somatic genetic events. While these cooperating driver mutations have been extensively studied in human and animal models, the true impact of CNAs in cancer initiation, development, response to treatment and outcome remains mostly unknown due to the complex nature of these numerical and structural alterations. 

With an incidence varying between 25 and 80% [1,5], a better understanding of the role of CNA/CNV in cancer is key to identifying new vulnerabilities that can lead to the development of novel targeted therapeutic strategies. In this review, we will integrate clinical and fundamental observations of CNA/CNV in cancer, with a particular focus on haematological malignancies to highlight the uprising interest in understanding the impact of CNAs in tumourigenesis to develop innovative therapies. 

## 2. CNA/CNV Detection 

From early karyotyping to next-generation sequencing (NGS), the detection of CNA/CNV has vastly evolved in recent decades. Figure 1 shows the main techniques and key steps of this evolution; other approaches and their limitations have been extensively described elsewhere [11,12]. Compared to historical conventional cytogenetics, new detection tools have incrementally refined several features of CNAs such as the size, the specificity (at the gene, transcript or single nucleotide variation (SNV) level) and the depth (to distinguish between cancer and normal cells and assess intra-tumoural heterogeneity). Karyotyping and fluorescence in situ hybridisation (FISH) approaches are still commonly used for the diagnosis of many diseases, whether they are cancer-related or not, but are limited by depth. Towards the turn of the century, the development of array platforms such as comparative genomic hybridisation (CGH) and single nucleotide polymorphism-based microarrays (SNP-arrays) significantly advanced our ability to detect and precisely characterise CNAs, emphasising their incidence and complexity in cancer [13,14,15]. Polymerase chain reaction (PCR)-based detection methods, such as quantitative multiplex PCR of short fluorescent fragments (QMPSF) [16], multiplex amplifiable probe hybridisation (MAPH) [17], and multiplex ligation-dependent probe amplification (MLPA) [18], were subsequently developed and allowed CNA detection with 150 bp resolution. MLPA is commonly used in the clinic and allows for a cheap, fast, and precise detection of known CNAs. With the advent of NGS, new approaches and bioinformatic pipelines can now be used to detect CNAs and characterise them at the nucleotide level (as bulk or single cell), from DNA and RNA [19,20]. Furthermore, the development of optical genomic mapping (OGM) that uses chromosome band patterns from single DNA strands, assembled bioinformatically to identify genomic alterations at high resolution (de novo assembly: 500 bp), now offers efficient analysis of genetic alterations within a genome without sequencing [21]. Notably, OGM provides an effective tool to detect CNAs in cancer, as shown in haematological malignancies [22], and may represent a cost-effective option for genetic screening in future clinical settings [23]. Implementation of these new detection methods along with a plethora of novel bioinformatic tools has highlighted the role and incidence of CNAs in tumour development, maintenance and response to treatment, and in several cases, associated specific CNA/CNV with clinical outcomes, as further detailed below.

## 3. Types of CNA/CNV 

CNA/CNV is a general term describing a myriad of genomic/chromosomal aberrations that can be inter- and intra-chromosomal; the largest ones such as aneuploidy or chromosomal rearrangements resulting from duplication, deletion or chromothripsis are usually seen by conventional cytogenetic approaches.

Aneuploidy englobes gains and/or losses of whole chromosomes within a single cell, and can affect one single chromosome (as in monosomy and trisomy), several chromosomes (as in hypodiploidy or hyperdiploidy), and even lead to haploid (or ‘nearly’ haploid) or polyploid (as in triploidy or tetraploidy) karyotypes in some cases (Figure 2A,B) [24]. This large variety of aneuploidies, which can vary between tumours, within the same tumour type, or even within the same patient, has hindered our capacity to estimate their true impact on cancer development. Notably, some of these alterations have been associated with specific outcomes and are now integrated into clinical risk stratification, as discussed below. Aneuploidy is often, but not exclusively, the result of chromosomal instability (CIN) [25]. It can occur via various mechanisms including single mutations affecting genes encoding key regulators of chromosomal segregation or mitotic/meiotic process, including improper kinetochore–microtubule attachments (*AURKB* and *AURKC*), defective cohesin complex (*SMC1/3*, *RAD51*, *STAG2*) and defective spindle assembly checkpoint (*BUB1/3*, *MAD1/2*, *BUBR1*, *MPS1*) [26,27,28]. Importantly, CIN leads to ongoing chromosomal mis-segregation throughout cell progeny, resulting in tumour cell populations with different karyotypes, thus increasing intra-tumoural heterogeneity [29,30]. Notably, aneuploidy can trigger increased genomic instability (GIN) and CIN [31,32], with Passerini et al., showing that aneuploidy increases the frequency of pre-mitotic errors and DNA damage and enhanced sensitivity to replication stress, ultimately leading to genomic rearrangements. The reverse is also true as GIN induced by DNA mutations and epigenetic modifications (DNA methylation and histone modification) has been shown to lead to deletions/duplications [25,33]. In high hyperdiploid B-cell acute lymphoblastic leukaemia (HeH B-ALL, >50 chromosomes), chromosomal gain has been suggested to occur in various ways, including chromosome loss from tetraploid cells, abnormal mitosis linked to impaired condensin and downstream Aurora B kinase function, chromosomal gains resulting from CIN, as well as cell fusion and tripolar division with additional clonal selection over multiple generations [34,35,36]. To date, the exact mechanism and kinetics of the appearance of the hyperdiploid karyotype in HeH B-ALL are still debated, with some studies indicating that it may be an early event during tumourigenesis [37,38]. Moreover, emphasising a putative functional advantage of CNAs for tumourigenesis and during clonal selection, there is a lack of randomness of the chromosomes often gained in hyperdiploid B-ALL (gains of chromosome 4, 6, 10, 14, 17, 18, 21, and X) [39], many of which are retained in hypodiploid (HoH) B-ALL (retained chromosome X/Y, 14, 18, and 21) [40].

For several decades, cytogenetic approaches (karyotyping, FISH or CGH arrays) have also described chromosomal structures and chromosomal rearrangements associated with the amplification of specific DNA segments that can be released from the original chromosome, such as extrachromosomal DNA (ecDNA, also known as double minutes) and ring chromosomes, or remain integrated within the affected chromosome in homogeneously staining regions (HSR) (Figure 2C–G). In most cases, amplification of genomic regions due to these CNAs leads to an elevated expression of oncogenes. Among the best-known examples are the amplifications of *MYCN* and *MYC* oncogenes in neuroblastoma and acute myeloid leukaemia (AML), respectively, in double minutes or HSR structures [41,42]. These chromosomal alterations can result from several mechanisms that are not mutually exclusive, including breakage–fusion bridges (Figure 2C, first reported in 1941) [43] and the more recently described chromothripsis (Figure 2D) [44]. Chromothripsis is described as a one-off ‘catastrophic’ event that initiates with breakage–fusion bridge cycles resulting in a rearranged chromosome characterised by a distinctive pattern of deleted and amplified genomic fragments, in random order and orientation [44,45]. Chromothripsis is part of a larger family of chromosomal rearrangements, which also include chromoplexy and chromoanasynthesis. Major breakthroughs into the nature, origin and mechanisms of formation of these massively rearranged structures have been achieved with the implementation of next-generation sequencing approaches (reviewed in [46]). Chromothripsis has been shown to occur in numerous cancer types, including leukaemia, brain tumour and lung carcinoma [47]. A well-known example of chromothripsis in leukaemia is the intrachromosomal amplification of chromosome 21 (iAMP21), a rare subtype of childhood B-ALL associated with a high risk of relapse [48]; this rearrangement can also be found to occur in AML [49]. The mechanism of formation of iAMP21 in B-ALL has been elegantly revealed by Li et al. using a next-generation sequencing approach indicating a rearranged chromosome that alternates gains and losses of key genomic regions/genes of chromosome 21; in some instances, this whole process can be promoted by a predisposing Robertsonian translocation [50]. Chromothripsis has also been described in paediatric medulloblastoma and is associated with *TP53* mutations [45]; a functional link is also observed in AML [49]. Notably, it is commonly seen in the benign tumour, uterine leiomyoma, but does not seem to be sufficient to drive malignant transformation [51]. Although there is still much more to do to better understand chromothripsis, it is now well established as a significant contributor to cancer development. Additionally, chromothripsis is one of the mechanisms that can lead to the release of extrachromosomal material (Figure 2D). ecDNA usually involves double stranded breaks, replication and circularisation of small fragments that contain one or a few genes and exist separately to chromosomes (Figure 2E) [52]. These ecDNA are commonly found in cancer and a recent observation has shown that oncogene overexpression through ecDNA formation contributes to tumour heterogeneity and genome evolution during cancer progression [53].

Somatic intrachromosomal deletions and duplications ≥ 10 kb are observed in 87.5% of cancer [5]. Interestingly, deletion/duplication of specific genomic regions is commonly found as germline CNV (80–85% deletion, 10–15% duplication) [54]. Classically, deletions are suspected to contain gene or regulatory elements with tumour suppressor function whereas duplications/amplifications contain tumour promotor genes and oncogenes. A well-documented example of this is the amplification of the *MYCN* gene (located in 2p23-24) in approximately 20–30% of neuroblastoma cases [55,56]. *MYCN* amplification can exist either intrachromosomally as well as in ecDNA and can result in over 500-fold amplification in some cases [57]. Other key examples are amplification of *EGFR* (7p11.2) in about 40% of glioblastoma cases [58], *ERBB2/HER2* (17q12) in 15–20% of breast cancer cases [59] and *MET* (7q31) in 1–2% of metastatic colorectal cancer cases [60]. Amplifications of these oncogenes have been shown to have prognostic value, further discussed below. Notably, similar to the concept of oncogene addiction [61], the term ‘aneuploidy addiction’ has also been used to describe the nature of cancers that rely on this specific type of CNA [62]. Indeed, in their recent study, Girish et al. demonstrated that trisomy of chromosome arm 1q is associated with suppression of TP53 function through overexpression of *MDM4* (1q32.1) [62].

Overall, a large variety of CNAs/CNVs exist that arise through different mechanisms. Notably, similar to what has been described for small indels (insertions/deletions) and single nucleotide mutations [63], Steele et al. described signatures that can explain the CNA pattern of 97% of samples across 33 tumour types, providing a mechanistic explanation of how most of these large chromosomal rearrangements may arise in cancer [3]. These CNAs are clonally selected throughout the tumourigenic process; therefore, the kinetics of appearance (spatial and time-dependent), as well as their interaction with other genetic alterations (including with other CNAs within the same cell), dictates cellular fitness, tumourigenesis, response to treatment and relapse initiation.

## 4. Incidence and Clinical Impact of CNA/CNV in Cancer

Depending on the subtype of CNA/CNV, the dosage of one to thousands of genes or regulatory elements can be affected, and consequently modify key biological processes. For most of them, the true functional impact of these CNAs/CNVs remains elusive, but it is now well established that they can provide clinical insight. Notably, the type of CNA and the genomic regions implicated may strongly relate to tumour type, and some insights have been unveiled through studying the influence of somatic and germline CNV or genetic disorders on cancer incidence.

### 4.1. Somatic CNAs in Cancer

Independent of the techniques used to detect somatic CNAs, pan-cancer genomic studies have estimated the incidence of aneuploidy (60–90%), chromothripsis (50%), and other duplication/deletions (25–80%) in primary samples [1,2,3,5,64]. The amount, type and size of somatic CNAs can be highly variable between cancer [2,5]. An overview of the key CNAs observed in the most common solid tumours and in acute leukaemia is shown in Table 1. 

CNA burden (i.e., the ratio of total CNAs on the length of the whole genome) has been identified as a prognostic factor for recurrence and death [4]. Pan-cancer studies have also highlighted the impact of CNAs during tumour development, progression and therapeutic resistance, some of which are commonly deleted/amplified across cancer [11,48,65,66,67]. 

**Table 1 ijms-25-06815-t001:** Overview of the main CNAs affecting autosomes in cancer.

Cancer Type	CNAs (amp/del)	Frequency	Genes of Interest	Refs
** * Solid Tumour * **				
Breast cancer	+1q +8q +11q13 +17q −8p −16q	>50% 20–40% 17–51% 9–10% 45–55% >50%	*MYC* *CCND1* *ERBB2*	[68,69,70]
Prostate cancer	+8q +16p +7 −8p −13q −16q	30–35% 25–50% 25% 60–65% 50–55% 50–55%	*MYC* *EGFR* *NKX3-1* *RB1* *CDH1*	[71,72]
Lung cancer	+5p +1q +7p −8p −19p −9p	60–65% 55–60% 40–45% 45% 40–45% 40–45%	*TERT* *ARNT* *EGFR* *CCNE1* *CDKN2A/B*	[73,74]
Colorectal cancer	+7 +13q +20q −8p −17p −18	41–46% 56% 72% 50% 56% 61–66%	*EGFR, MET* *AURKA, ASXL1, HNF4A* *TP53* *CADH7*	[75,76,77]
Skin cancer	+7 +6p +8q −6q −9 −10	50–55% 20–50% 5–40% 30–50% 20–50% 5–50%	*EGFR* *MYC* *CDKN2A/B* *PTEN*	[78,79,80]
** * Acute leukaemia * **				
Adult AML	+8 +21 +22 −5/5q −7/7q −17/17p	10–15% 4–6% 4–6% 10–15% 6–7% 5–6%	*MYC* *ERG* *EZH2* *TP53*	[81,82]
Paediatric AML	+8 +21 +19 −5/5q −7/7q	10–14% 7–10% 5% 1.2% 4–5%	*MYC* *ERG* *EZH2*	[81,83]
Adult B-ALL	+21 −9p21 −7p12 −9p13	11% 25–40% 30–40% 20–40%	*RUNX1, DYRK1A* *CDKN2A/B* *IKZF1* *PAX5*	[84,85]
Paediatric B-ALL	+21 +6 +14 −9p21 −9p13 −7p12	27% (100% HeH) 90% in HeH 90% in HeH 30–40% 20–30% 15–20%	*RUNX1, DYRK1A* *CDKN2A/B* *PAX5* *IKZF1*	[38,86,87]

Data were extracted from references and using cBioportal [88] and the Mitelman database (available at https://mitelmandatabase.isb-cgc.org/) [89,90]. Lists of CNAs are not exhaustive and represent common alterations seen in the selected tumour type.

Somatic CNAs affecting well-known oncogenes and tumour suppressor genes such as *MYC* (located in 8q24), *EGFR* (7p11.2), *TP53* (17q13.1), *RB1* (13q14.2), and *CDKN2A/B* (9p21.3) are commonly seen across cancer. A recent study using computational analysis of 10,884 patients across 33 cancer types reported that the gain of human ch1q has prognostic value across multiple cancers [62]. In solid tumours, genomic amplification (amplicons) of *EGFR*, *MET*, *CCND1*, or *ERBB2* have been associated with inferior survival rates and have correlated with resistance to targeted therapies [91,92,93,94,95]. In prostate cancer, a recent study conducted by Alfahed et al. identified several markers associated with disease stage, Gleason grade group features and progression-free survival, confirming earlier observations [67,71]. They also showed that the combination of seven markers can differentiate patients with localised and advanced stages of prostate cancer. Another important aspect of screening CNAs in cancer patient samples is the potential to identify if CNAs infer sensitivity or cytotoxicity to therapeutic agents, as recently shown in melanoma and neuroblastoma [96,97]. Wong et al. correlated the gain of *PPM1D*/*GNA13* genes or loss of *CBL*/*DNMT3A* genes with enhanced cytotoxicity, whilst gains of *PI3K* or *STAT* family genes sensitise neuroblastoma cells to cell cycle inhibitors [97]. Moreover, using the cancer proteome atlas, a predictive link between CNAs and phosphorylation changes has been discovered, providing a new potential readout to assess sensitivity to kinase inhibition in cancer [98]. 

At the chromosomal arm level, amplifications of 5p, 8q and 7p are among the most frequent CNAs seen in solid tumours and are often associated with 5q, 8p and 7q deletions through the formation of isochromosomes. This is observed in ≈25% of solid tumours and the most frequent are i3q, i5p, i8q and i20q, with i8q being one of the most broadly acquired isochromosomes [99]. i8q or 8p deletions have also been associated with disease progression and poorer prognosis in breast, colorectal and prostate cancer [7,100,101]. In breast cancer, Cai et al. have shown that 8p deletions confer a loss of heterozygosity (LOH) phenotype, which results in alterations to lipid metabolism with an increase in drug resistance and tumour invasiveness [7]. Interestingly, while chromosome 7 is mostly gained in solid tumours, it is frequently lost in haematological malignancies (Table 1). This reflects the tissue specificity of CNAs in cancer and is most likely due to the presence of key oncogenes in solid tumours (*EGFR*, *MET* and *BRAF*), and well-known tumour suppressor genes in blood cancers (*CUX1* and *IKZF1*).

Paediatric B-ALL is a well-known example of how treatments have been adapted using risk-stratification based on genetic alterations, including CNAs, to successfully improve outcomes. HeH B-ALL is the most common subtype of childhood B-ALL (25–30% of cases) and has a good prognosis overall [102]. The presence of double (+4, +10) or triple (+4, +10, +17) trisomy has also been shown to confer better outcomes [103,104]. Furthermore, with regard to drug sensitivity, ex vivo pharmacotypic profiling performed in a recent study revealed differential sensitivity to asparaginase, cytarabine and thiopurines in paediatric HeH B-ALL. Several chromosome-specific associations were identified, including the gain of chromosomes 7 and 9 with asparaginase resistance, the gain of chromosomes 16 and 17 with asparaginase sensitivity, and the gain of chromosomes 14 and 17 with mercaptopurine sensitivity [105]. 

At least two other subgroups are strictly defined by CNAs in B-ALL: HoH and iAMP21, both of which have poorer outcomes [102]. Inferior outcomes and drug resistance have also been associated with deletion of the *IKZF1* gene (7p12) in paediatric B-ALL but its impact on the outcome of adult B-ALL remains controversial; *IKZF1* encodes a critical regulator of lymphoid differentiation [85,86,106]. Notably, a recent study has recently refined the prognostic value of *IKZF1* alterations in a large cohort of paediatric B-ALL, demonstrating that MRD-positivity in patients with *IKZF1*^plus^ signature (i.e., *IKZF1* deletion plus other co-occurring deletions such as *CDKN2A*, *PAX5* or *PAR1*) are associated with a very-poor prognosis [107]. CNAs also hold significant prognostic value in myeloid leukaemia. Deletions of the long arm of chromosome 7 (−7q) occur in up to 33% of AML cases [108,109]. In paediatric AML, monosomy 7 or 7q deletions are associated with a poor prognosis [108,110]; 5q deletions also result in inferior survival outcomes in AML and are sometimes observed together with 7q deletions [111]. Other common CNAs seen in leukaemia are gain of chromosome X in HeH B-ALL and loss of chromosome Y in AML but their role in leukaemogenesis and prognostic value remains unclear [81,87]. Notably, some CNAs seen in childhood acute leukaemia (+21 in B-ALL) are less frequently acquired in adult malignancies and vice versa (−5/5q in adult AML), emphasising the time and spatial sensitivity to CNAs for cellular transformation.

Altogether, these data illustrate the impact of integrating CNAs as prognostic factors to inform risk stratification and further explore sensitivity to conventional treatments

### 4.2. Constitutional CNV and Cancer 

The impact of CNAs on cancer has also been unveiled by studying the phenotypes associated with inheritable and constitutional CNV-driven syndromes. Interestingly, studies assessing CNAs amongst healthy individuals revealed that 5–12% of the genome is subject to these genomic variations [112,113]. This includes large-scale differences of several kilobases or greater, which for the most part do not result in disease phenotypes [114]. In line with these observations, Zarrei et al. highlighted that about 100 genes commonly implicated in CNV could be deleted without being associated with phenotypic differences [113]. In 2010, Conrad et al. referenced 30 loci with a potential role in the predisposition to specific diseases such as hepatitis B, schizophrenia, Crohn’s disease and cancer [115]. Several germline or constitutional CNVs have been shown to increase the risk of cancer and affect development, progression, response to treatment, relapse and outcomes [116] (Table 2). 

Among constitutional aneuploidies, individuals with Down syndrome (DS, trisomy 21), Turner syndrome (45,X) or Klinefelter syndrome (47,XXY) face a specific pattern of cancer incidence compared to the general population, with both an increased and decreased risk of developing tumours depending on time and tissue/organ [117,118,119,120,121,122,123]. Exemplifying this tumourigenic pattern, children with DS, but not adults, are at a significantly higher risk of developing both myeloid leukaemia (ML-DS, with features of megakaryoblastic leukaemia, 150-fold increased) and ALL (DS-ALL, 20-fold increased) compared to children without DS, while they have a reduced frequency of solid tumours apart from testicular germ cell tumour (reviewed in [117]). ML-DS is preceded by a pre-leukaemic transient abnormal myelopoiesis (TAM) that initiates during foetal life and often regresses spontaneously shortly after birth. Indicative of the impact of trisomy 21 in tumour initiation, Roberts et al. showed that nearly all neonates with DS exhibit perturbed haematopoiesis characterised by higher haematocrit and dyserythropoiesis, lower platelet count and several leukocyte abnormalities. More than 95% of cases had blasts in the peripheral blood and 20–30% of them developed TAM through the acquisition of a mutation in the *GATA1* gene [124]; the progression from TAM to ML-DS requires additional somatic alterations [117,125,126]. While the event-free survival is favourable and is approaching 90% in ML-DS, a recent international retrospective study showed that survival of relapsed/refractory ML-DS remains below 25% [127]. Children with DS are also predisposed to develop ALL with inferior outcomes compared to non-DS children due to increased incidence of relapse, death in induction and death in remission [128,129,130]. The genomic landscape of DS-ALL has been recently refined, emphasising new potential targetable mechanisms of oncogenic cooperation with trisomy 21 [131]. Of note, the pattern of gain of chromosome 21 as a somatic event in non-DS leukaemia (ranging from 4–34% in blood cancer in general) is further indicative of the pro-tumourigenic role associated with extra copies of this chromosome [132]. Other less frequent constitutional aneuploidies affecting autosomes such as trisomy 13 and trisomy 18 have also been shown to have an increased incidence of cancer; however, their prognostic value is not known due to the rarity of these CNV-driven cancers [133,134,135]. Constitutional trisomy 8 mosaicism has been shown to predispose to myeloid leukaemia in individual cases [136], which may correlate with somatic +8 being observed in 10–15% of AML cases [81,83].

Constitutional gain or loss of sex chromosome X has also been associated with variable incidence of cancer (Table 2). Several reports have shown that females with Turner syndrome (45,X) have a decreased incidence of breast cancer, although this remains contentious [119,120,121]. Notably, a gain of chromosome X in Klinefelter syndrome (47,XXY) leads to a higher rate of developing breast cancer, together with an increased incidence of non-Hodgkin lymphoma and leukaemia compared to the general population [123]. To date, there is no clear evidence of an increased incidence of breast cancer in constitutional trisomy/tetrasomy X syndromes; however, it has been speculated that this may be due to the fact that the additional X chromosomes are inactivated [137]. Strikingly, somatic gain of chromosome X, as well as defects in X inactivation, has been associated with more aggressive disease and poorer prognosis in breast cancer [138]. Extra copies of chromosome X have also been reported in pancreatic cancer as well as in specific subtypes of childhood ALL [87,139,140], but to date, its role in tumourigenesis remains unclear. Additionally, a large study of 2561 British patients with constitutional autosomal deletions of chromosomal arms, confirmed the known association between deletion 11p and 13q with Wilms tumour and retinoblastoma, respectively, and identified that deletions of the genomic region 11q24 may predispose to anogenital cancer, in line with the recurrent somatic alterations described in these solid tumours [141,142,143]. Notably, several studies have reported cancer cases with other constitutional CNV such as Prader–Willi syndrome (deletion 15q11.2-q13) [144], DiGeorge syndrome (deletion 22q11.2) [145], Cri du Chat syndrome (deletion 5p) [146], and Williams–Beuren syndrome (deletion 7q11.23) [147], but their true incidence and any potential link with clinical outcomes remains unclear.

Altogether, whether they are pro- or anti-tumourigenic, the role of CNA/CNV in tumour development and response to treatment is now well established. This is in line with the increasing amount of pharmacogenomic studies showing that germline alterations not only predispose to cancer but also modify treatment outcomes between patients that have a similar tumour type [148]. However, the exact role of most CNA/CNV remains elusive and is likely to depend on the cellular context, temporal acquisition and cooperation with other genetic alterations (including other CNAs). Due to their high incidence, a better understanding of the intimate link between CNAs and tumourigenesis would provide new opportunities to prevent and improve outcomes for individuals with cancer.

**Table 2 ijms-25-06815-t002:** Common CNV-related disorders associated with cancer.

CNV-Driven Syndromes (Incidence)	Type of CNV	Higher Incidence of Cancer	Lower Incidence of Cancer	Refs
Down syndrome (1/700–800)	Trisomy 21	ALL, TAM/ML-DS, testicular germ cell tumour	Solid tumours	[117,118]
Edwards syndrome (1/6000)	Trisomy 18	Hepatoblastoma, Wilms tumour, benign cardiac tumours *	N/A	[134,135]
Patau syndrome (1/10,000–20,000)	Trisomy 13	Embryonic tumours, malignant germ cell tumours, leukaemia, carcinoma, brain cancer and sarcoma	N/A	[133]
Turner syndrome (1/1200–2500, female births)	Monosomy X (45,X)	Benign/malignant skin neoplasm and cancers, brain tumours, bladder and urethral cancer, colorectal cancer	Breast cancer	[119,120,121]
Klinefelter syndrome (1.72/1000 male births)	Extra X (47,XXY)	Breast cancer, lung cancer, germ cell tumours, non-Hodgkin lymphoma, ALL and myeloid leukaemia	Solid tumours, prostate cancer	[120,122,123]
Warkany syndrome 2 (1/25,000–50,000)	Mosaic Trisomy 8	Myeloid leukaemia *	N/A	[136]
Prader–Willi syndrome (1/15,000–30,000)	Deletion 15q11.2-q13	ALL and myeloid leukaemia, biliary cancer, melanoma, hemangiopericytoma, adenocarcinoma, colon cancer *	N/A	[144,149]
11p deletion/WAGR syndrome (<1/500,000)	Deletion 11p13	Wilms tumour	N/A	[143]
13q deletion syndrome (very rare)	Deletion 13q14	Retinoblastoma	N/A	[143]
11q24 deletion syndrome (very rare)	Deletion 11q24	Anogenital cancer	N/A	[143]

* based on case studies. N/A = not applicable.

## 5. Overview of the Mechanisms Disturbed by CNA/CNV in Cancer

As CNA/CNV perturb the dosage of genes and regulatory elements (promotors, enhancers, microRNA), it is viewed that they ultimately alter the expression of hundreds or thousands of genes, not only in the chromosomal regions amplified or deleted but also genome-wide, thus impacting key biological functions such as transcription, splicing machinery, DNA methylation, histone modification and cell metabolism, among others. In cancer cells, it has been proposed that CNAs promote tumourigenesis by providing the cellular fitness required for transformation, depending on the balance of tumour-suppressive and oncogenic function of the genes included within the amplified/deleted genomic region. Whether these alterations are direct oncogenic drivers or passively promote genetic instability to facilitate tumourigenesis remains elusive. Thus, understanding the impact of this CNA/CNV and identifying the dosage-sensitive genes may provide a detailed network of cancer-promoting/targetable pathways. However, this is a complex task as (1) there are a limited amount of clinically relevant models that can be used to assess CNA/CNV function, (2) not all genes/regulatory elements included in the CNA/CNV are dosage sensitive and so may not be targetable, (3) tumour heterogeneity (cohabitation of distinct genetic phenotypic clones) may impede the discovery of dosage-sensitive mechanisms, (4) CNAs may cooperate with other genetic alterations (including other CNAs) that could be tissue-specific and may be modulated by the tumour microenvironment. As an example of the latter, xenograft models have revealed that CNAs can provide cellular fitness and facilitate clonal selection in an exogenous murine environment [150]. In this section, we will present selected examples of models and experimental approaches used to dissect the role of CNAs in cancer, drawing a snapshot of the key mechanisms they alter and exposing the challenges that remain to fully understand their role.

### 5.1. Identifying Dosage Sensitive Genes

The number of genes/regulatory elements included within the CNAs does not always correlate with tumourigenicity, as not all of them are expressed or active in a specific cell type, at a specific time (ontogenic and/or cellular state, therapeutic response and resistance) or in a specific clinical condition (healthy, chronic disease, cancer progression). It is also tempting to speculate that cellular function affected by dosage sensitivity may be dependent upon the precise ‘number of copies’ of key genes/regulatory elements. Apart from the CNAs that affect the expression of well-known tumour-associated genes (such as *MYC*, *TP53*, *RB1* and *CDKN2A*/*B*), the true impact of many genes affected by CNAs and whether these genes are dosage-sensitive remains unknown. Genomic analyses aiming at refining minimal regions that are deleted or amplified between different specimens have facilitated the identification of such genes.

According to the ‘two-hit’ Knudson hypothesis, the minimal genomic region deleted in 11p13 deletion/WAGR syndrome led to the identification of one of the first tumour suppressor genes *WT1* [151,152], a gene that encodes a zinc finger transcription factor known to control transcription via its direct interaction with DNA, DNA hydroxymethylation and RNA metabolism. Interestingly, WT1 has been shown to regulate many cellular processes, such as cell proliferation, survival and differentiation, as well as tissue-specific processes such as mesenchymal–epithelial transition in the nephron and epithelial–mesenchymal transition in the developing heart [153]. These contrasting roles are exemplified by the observation that *WT1* is now also considered as an oncogene in several cancers including breast cancer, glioblastoma, pancreatic cancer and AML. In the latter, *WT1* is overexpressed in more than 80% of cases, has been associated with resistance to therapy, increased rate of relapse and inferior overall survival, and can be used as a biomarker for the detection of minimal residual disease [154]. Moreover, amplification of the tyrosine kinase receptors *HER2/ERBB2*, *MET* and *EGFR* has been observed in many solid tumours (breast, lung and brain, among others), driving tumourigenesis through the uncontrolled activation of downstream signalling (such as PI3K/PTEN/mTOR and RAS/MAPK pathways) leading to growth, proliferation and survival; amplifications of these genes are also commonly associated with metastasis and therapy resistance [91,155,156,157].

One of the best examples of the impact of large CNA/CNV in tumourigenesis lies in understanding the critical balance of tumour suppressor genes and tumour promotor dosage-sensitive genes located on chromosome 21. Figure 3 proposes several *scenarios* that may explain the preventative and predisposing/promoting role of trisomy 21. This small chromosome 21 is gained more often than any other chromosome in haematological malignancies, which is in contrast to solid tumours [2] (Table 1). Strikingly, the incidence of complete or partial gain of chromosome 21 reaches 25–30% in paediatric B-ALL and 30–35% in acute megakaryoblastic leukaemia (AMKL), and the observation that individuals with DS are predisposed to B-ALL and AMKL indicates that both the B-cell and megakaryocytic lineages are strongly sensitive to the increased dosage of chromosome 21 genes during childhood [132]. To investigate this dichotomy, several groups have used murine models of DS harbouring various sizes of trisomy 21 to identify the chromosome 21 genes that are implicated (reviewed in [132]). It has been confirmed that trisomy 21 promotes leukaemia development [158,159,160], but also prevents solid tumour formation, the latter being linked to the impact of trisomy 21 on neural development, angiogenesis and metabolism [118,161,162,163]. Interestingly, this confirms that several dosage-sensitive genes can either promote or suppress tumourigenesis and that several dosage-sensitive chromosome 21 genes cooperate to drive these phenotypes. For instance, it has been shown that trisomy of the chromosome 21 genes *DYRK1A* plus *RCAN1* decreases solid tumour growth through the additive inhibition of the Calcineurin/NFAT pathway [161]. Using a similar approach, Reynolds et al. demonstrated that trisomy of *JAM-B*, *PTTG1IP, ADAMTS1* and *ERG* (a well-known oncogene in prostate cancer and leukaemia) prevents tumour growth by inhibiting neo-angiogenesis [162]. In line with tissue specificity, several studies have shown that increased dosage and expression of *ERG*, *ETS2*, *HMGN1*, *CHAF1B* and *DYRK1A*, as well as disequilibrium in transcripts of the chromosome 21 gene *RUNX1* induced by trisomy 21, promote leukaemia development [158,159,164,165,166,167]. Interestingly, DYRK1A has been shown to modulate the activity of many effectors with oncogenic and tumour suppressive functions (STAT3, NOTCH1, FOXO1, Cyclin D1/D3, Rb1, TP53, P21) in a tissue-dependent manner, altering several signalling pathways through distinct mechanisms, regulating the cellular fitness, metabolism, DNA repair, cell division, proliferation/growth and survival in different cancers [168].

### 5.2. Role of large CNA/CNV and Aneuploidy in Cancer Development

Characterisation of single dosage-sensitive genes has the potential to lead to novel targeted therapies but does not offer the opportunity to understand the role of CNA/CNV as a whole, with potentially hundreds or thousands of genes affected by dosage imbalances. Historically, animal models harbouring aneuploidies have been used to better understand the impact of chromosomal deletion and amplification in specific syndromes such as Prader–Willi syndrome, Williams–Beuren syndrome or DS (reviewed in [169]). While crucial mechanisms have been identified using these models, the location of the chromosomal regions syntenic to the human chromosomes within the murine genome may also be a confounding factor. As an example, syntenic regions of human chromosome 13 (Hsa13) are spread over six murine chromosomes and Hsa21 over three murine chromosomes: Mm10, Mm16 and Mm17. Therefore, trisomy of Mm16 (as in Ts65Dn, the most commonly used model to study DS) does not completely recapitulate human trisomy 21, as it is disomic for Mm10 and Mm17 and has additional Mm16 trisomic genes that are not syntenic to Hsa21 [170]. To bypass these limitations, new models of DS have been established. This includes transgenic models triplicated for Mm10, Mm16 and Mm17 [171], as well as the Tc1 and TcMAC21 models, which were developed using microcell mediated chromosome transfer [172], that span 70–75% of the protein-coding Hsa21 genes [173,174]. As mentioned above, many of these models have been used to dissect the role of trisomy 21 in promoting leukaemia and preventing solid tumour development.

Whereas understanding the impact of aneuploidy remains challenging in transgenic mice, several key observations have been obtained using cellular models. In mouse embryonic fibroblasts (MEF), it has been shown that single gains of murine chromosome 1, 13, 16 and 19 decrease tumourigenicity, in part by promoting premature growth arrest, while a gain of murine chromosome 2 increased proliferation; reduction in tumourigenicity due to specific trisomies was also reported in the colorectal cell line HCT116 [175]. However, in another study, Vasudevan et al. demonstrated that trisomy of chromosome 5 specifically promotes tumour progression and invasiveness of HCT116 cells [176]. Contrasting results are also observed in MEF cells that have reduced levels of the centromere-linked motor protein E (CENP-E^+/−^), where aneuploidy can both promote or inhibit transformation in vitro, and tumourigenesis in vivo in a tissue-dependent manner (lymphoma versus liver cancer, respectively) [177]. Other roles known to be affected by CNAs in cancer are linked to cell metabolism (reactive oxygen species levels, glycolysis, mitochondrial activity and endoplasmic reticulum stress) and modulation of the immune environment (reviewed in [178]) [179,180]. This evidence supports the dichotomic role of CNAs in cancer and re-emphasises the complexity of understanding their true impact on cancer development. 

With the advent of gene editing technologies such as Transcription Activator-Like Effector Nucleases (TALEN) and Clustered Regularly Interspaced Short Palindromic Repeats (CRISPR), new approaches have been designed to reproduce and dissect the role of CNA/CNV in debilitating diseases and cancer. For example, Tai et al. applied single-guide CRISPR/Cas targeting of repetitive elements (SCORE), a CRISPR-based approach that targets the homologous sites in flanking segmental duplications, to create microdeletion (15q13.3) and microduplication (16q11.2) syndromes [181]. In a model of breast cancer, Cai et al. used TALEN to model chromosome 8p LOH by non-homologous end-joining in MCF10A mammary epithelial cells (33 Mb deletion, downregulation of 81 protein-coding genes) and showed that alterations in lipid metabolism result in increased drug resistance and tumour invasiveness [7]. TALEN and CRISPR technology have also been used to restore gene dosage in trisomic samples. Focusing on DS and leukaemia, Banno et al. deleted a 4 Mb region of chromosome 21 in DS induced pluripotent stem (iPS) cells to show that the transcription factors ERG, RUNX1 and ETS2 play a role in TAM/ML-DS [182]. Alternative applications of CRISPR and TALEN technologies have allowed for the elimination and silencing of the extra chromosome 21 in DS iPS cells by inducing multiple DNA strand breaks or by integrating long non-coding RNA X-inactive specific transcript into the extra copy of chromosome 21, respectively [183,184]. Recently, CRISPR technology has allowed for the development of elegant approaches such as molecular alteration of chromosomes with engineered tandem elements (MACHETE) and restoring disomy in aneuploid cells using CRISPR targeting (ReDACT) to model CNAs in cancer [62,185]. MACHETE provides the opportunity to efficiently delete large chromosomal segments and has been used to show that homozygous deletion of the type I interferon cluster, seen in more than 10% of cancer, promotes metastasis by mediating immune evasion in a syngeneic murine model of pancreatic cancer [185]. Girish et al. showed that a gain of chromosome 1q associated with increased *MDM4* expression and suppression of TP53 signalling is an early event in cancer development and developed ReDACT to demonstrate that loss of trisomy 1q blocked growth and prevented malignant transformation of cancer cell lines in vitro and in vivo [62]. This system has been used for several other CNA/aneuploidy in different cellular contexts to confirm that these chromosomal alterations modulate cellular fitness and draw on the concept of ‘aneuploidy/CNA addiction’ in cancer. Overall, these selected examples show that TALEN/CRISPR technologies have been instrumental in dissecting the roles and mechanisms induced by CNAs in cancer.

Altogether, this evidence confirms that CNAs are key drivers of tumourigenesis with functions dependent upon multiple factors including the size (number of tumour promotor/suppressor genes/regulatory elements), the chromosome or genomic region affected, the cell type/tissue, tumour stage, clonal heterogeneity and the impact of the microenvironment. Despite increasing knowledge in this area of research, the true role of CNA/CNV remains controversial in some instances, which exemplifies the complexity of dissecting their role in cancer, as opposed to gene mutations or translocations. Further investigations are warranted and will require the development of innovative approaches to unlock new mechanisms of cancer development and identify novel therapeutic targets. 

## 6. Targeting CNA/CNV to Improve Outcomes

Due to their incidence in cancer and dosage dependency, therapeutically targeting CNAs/CNVs to restore disomy represents an attractive opportunity to improve patient outcomes. Whether this can be achieved directly by targeting the key dosage-sensitive genes/regulatory elements or the mechanisms they perturb, or by ‘unbalancing’ the cellular fitness provided by CNAs to disturb tumour homeostasis, is a promising therapeutic avenue. 

While alternative strategies are still needed to target the ‘undruggable’ *MYC* oncogene (overexpressed in about 70% of all cancers), targeting tyrosine kinases amplified and overexpressed in specific cancers has shown some success. Perhaps one of the best examples is therapy targeting *HER2/ERBB2* amplifications in breast cancer. In recent decades, new targeted agents have been developed, which are now used in the clinic to treat patients with breast cancer such as monoclonal antibodies (trastuzumab, pertuzumab, margetuximab) and tyrosine kinase inhibitors (afatinib, lapatinib, neratinib, pyrotinib and tucatinib) [186,187]. Other well-known examples of tyrosine kinase inhibitors and monoclonal antibodies proven to be of clinical use are those targeting c-MET (in lung cancer), EGFR (in breast, lung, gastric and pancreatic cancer), and FLT3 (acute leukaemia). However, various mechanisms of resistance can develop by the tumour cells themselves (e.g., gene amplification, mutations and clonal selection) or by the stroma (protective effects) to bypass the efficacy of these agents, thus emphasising the need for alternative strategies to improve long-term outcomes. 

Through the identification of new dosage-sensitive genes in children with DS and leukaemia, several groups have shown that targeting the product of the chromosome 21 genes *HMGN1* and *DYRK1A* may be broadly applicable to many blood cancers and potentially solid tumours [158,159,165]. Indeed, increased dosage of the high mobility group nucleosome-binding protein N1 (HMGN1) has been shown to upregulate B-cell-specific transcriptional signatures in DS-ALL as well as subtypes of B-ALL harbouring gain of chromosome 21, by antagonising the PRC2 complex (reduced H3K27me3 marks) and increasing chromatin accessibility (increased H3K27ac marks) [159]; notably, HMGN1 has also been shown to cooperate with the transcript fusion AML1-ETO9a to promote AML [188]. Restoring chromatin accessibility using inhibitors of histone demethylase (GSK-J4) or histone acetyl transferase (C646) has shown promising in vitro and in vivo results for both B-ALL and AML [159,188,189]. Another attractive target encoded by chromosome 21 is the dual-specificity tyrosine-regulated kinase DYRK1A. We and others have shown that trisomy of *DYRK1A* contributes to the development of both ML-DS and DS-ALL, and that inhibition of its kinase activity or of its targets (STAT3, FOXO1, Cyclin D3) decreases growth and survival in vitro and prolongs in vivo survival [165,190]. Results from in vitro drug combination testing are promising, and future directions should consider clinical evaluation in early phase trials for leukaemia, as well as for tumours with poor prognosis, such as glioblastoma or head and neck cancer, where DYRK1A has also been implicated [191,192]. Finally, a recent study from Memon et al. demonstrated that CNAs modulate signalling pathway activity in many cancers and can be used as a predictor of sensitivity to kinase inhibitors, suggesting that disturbing this refined and balanced activity may help to design more adapted and less toxic therapies [98]. This ‘kinase addiction’ phenomenon indicates that the use of biologically informed inhibitors has the potential to improve outcomes for patients with CNA-driven cancer. Interestingly, a recent study also demonstrated that aneuploid cancer cells are sensitive to spindle assembly checkpoint inhibitors due to their intrinsically perturbed activity of the kinesin KIF18A [193].

Further investigations are warranted to assess the functional output and crosstalk between CNA/CNV as well as with other somatic alterations within the individual cancer cell, between several clones, and with the tumour microenvironment. A clearer understanding of these mechanisms will allow the identification of biomarkers to develop new monitoring tools to better follow response to treatments and of an “Achilles heel” to design novel effective combination therapies to decrease treatment toxicity, prevent relapse and improve long-term survival.

## 7. Future Directions

CNAs/CNVs constitute a considerable genomic phenomenon that not only affects the primary target (i.e., the chromosome) but also creates several imbalances at the genomic, transcriptomic and proteomic levels. Thus, CNAs/CNVs may have huge consequences on cellular biology by affecting intracellular signalling, epigenetic mechanisms, metabolism, immune response, and therapy resistance among others, which ultimately result in providing cellular fitness to tumour cells. While they are one of the most known and frequent alterations seen in cancer cells, we are only just beginning to understand their true function. Clinically, CNAs/CNVs can be informative with regard to cancer predisposition, cancer development and progression, response to treatment and resistance, and long-term outcomes. Recent observations have emphasised that manipulation of CNAs strongly affects tumour homeostasis, suggesting that we have opened Pandora’s box regarding new opportunities to better understand tumourigenesis, identify new actionable targets and integrate dosage-dependent therapies into the clinic.

## Figures and Tables

**Figure 1 ijms-25-06815-f001:**
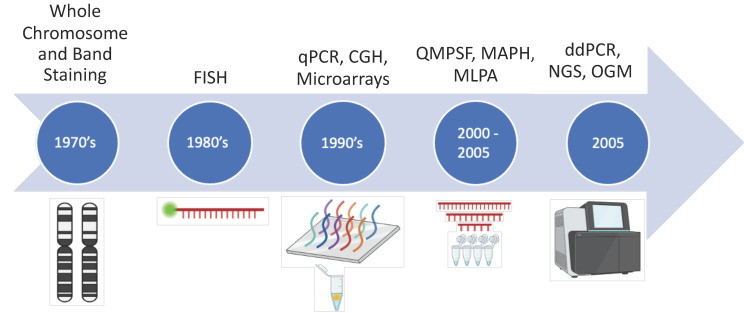
Main methods of detection of copy number alterations/variations. Timeline of the development of methods for detection of CNA/CNV, spanning from the 1970s to the present. Fluorescence in situ hybridisation (FISH), comparative genomic hybridisation (CGH), quantitative polymerase chain reaction (qPCR), quantitative multiplex PCR of short fluorescent fragments (QMPSF), multiplex amplifiable probe hybridisation (MAPH), multiplex ligation-dependent probe amplification (MLPA), digital droplet PCR (ddPCR), next-generation sequencing (NGS) and optical genome mapping (OGM). Created with BioRender.com (accessed on 21 May 2024).

**Figure 2 ijms-25-06815-f002:**
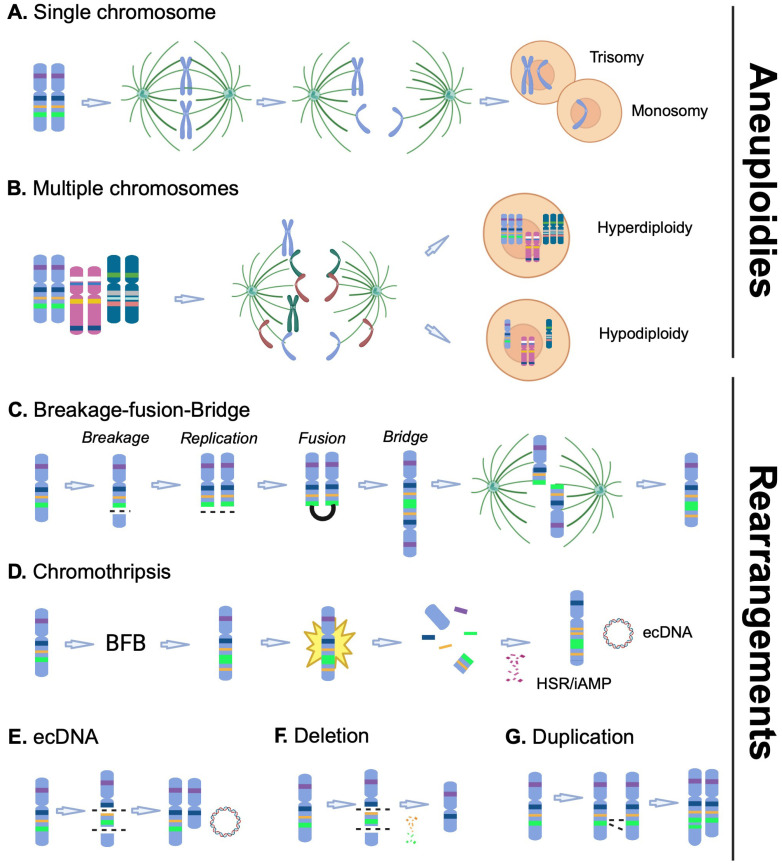
Mechanisms of formation of copy number alterations. Aneuploidies: (**A**) Chromosomes align during metaphase, followed by mis-segregation at anaphase that may lead to gain (trisomy) or loss (monosomy) of one chromosome. (**B**) Hypothetical mechanisms leading to hyperdiploidy and hypodiploidy due to mis-segregation of several chromosomes during anaphase. Rearrangements: (**C**) Amplification of genomic regions by a breakage–fusion bridge (BFB); several sequential BFB cycles can happen. (**D**) Chromothripsis is a catastrophic event that usually starts with BFB cycle(s), followed by the ‘explosion’ of an unstable rearranged chromosome and reintegration of DNA fragments prior to stabilisation. This process can lead to homogeneously staining regions/intrachromosomal amplification (HSR/iAMP), ring chromosomes and extrachromosomal DNA (ecDNA). (**E**) Extrachromosomal circular DNA occurs when a DNA fragment is excised and circularised; or following chromosomal rearrangements such as chromothripsis. (**F**) In deletions, genomic regions are lost mostly due to DNA double-strand breaks and non-homologous repair. (**G**) Duplication results from the exchange of homologous genomic segments between sister chromosomes. Dotted line represents DNA-strand breaks. Created with BioRender.com (accessed on 21 May 2024).

**Figure 3 ijms-25-06815-f003:**
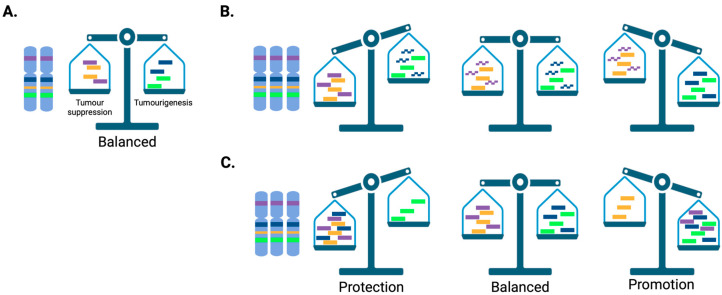
Dosage sensitivity and tumourigenesis. (**A**) Disomic cells have the right dosage of tumour-suppressing and tumour-promoting genes (or regulatory elements) to maintain cellular homeostasis. (**B**,**C**) Two examples of trisomic cells with different scenarios of genes/regulatory elements expressed (solid) versus not expressed (dashed) that may protect, not affect (balanced) or promote tumourigenesis in different tissues. Created with BioRender.com (accessed on 21 May 2024).

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
