# Peer review of "Insights into the Clinical, Biological and Therapeutic Impact of Copy Number Alteration in Cancer"

_ijms, 2024, doi:10.3390/ijms25136815_

Round 1

Reviewer 1 Report

Comments and Suggestions for Authors

The manuscript is well organized, data is presented very well, no major issues identified.

  1. In abstract, line 22, one word is missing, so the sentences seems incomplete.....integrating innovative genetic engineering strategies to ...what?
  2. In section 3, line198, the percentage of MET in colorectal cancer with reference 57 is mentioned as 15-20%, but the authors have observed/concluded the percentage of 1-2%, previous other studies have suggested 9-18%. Please explain the difference
  3. The Mitelman database mentioned in table 1 (page, 6, line 234) is not included in the reference section and I am unable to open this website
  4. Line 242, gene names are in different formats, for information only to reformat
  5. In section 4, many gene names need to be Italicized, this is a general rule and need to be followed throughout the manuscript
  6. Line 305, one reference is mentioned as recent but is from 2015, better delete recently from the sentence, it is a decade old reference
  7. Line 307, the year mentioned as 2009 does not match with reference year 2010.....Nature, 2010, 464......
  8. Line 456, Rb1 is mentioned as Rb, need to fix typo.

Author Response

Comments from Reviewer 1:

The manuscript is well organized, data is presented very well, no major issues identified.

1. In abstract, line 22, one word is missing, so the sentences seems incomplete.....integrating innovative genetic engineering strategies to ...what?

We agree and thank the reviewer for identifying this mistake. We have deleted the word ‘to’ in this context and modified the sentence as follows: “…and discuss integration of innovative genetic engineering strategies, to highlight the potential …”

2. In section 3, line198, the percentage of MET in colorectal cancer with reference 57 is mentioned as 15-20%, but the authors have observed/concluded the percentage of 1-2%, previous other studies have suggested 9-18%. Please explain the difference

We apologize for this mistake. The correct percentage is indeed 1-2% in metastatic colorectal cancer (mCRC) as per reference 60. We have corrected this in the revised manuscript (now on line 186, page 10).

3. The Mitelman database mentioned in table 1 (page, 6, line 234) is not included in the reference section and I am unable to open this website

We thank the reviewer for identifying this. We have now updated the link (https://mitelmandatabase.isb-cgc.org) and added the corresponding reference 89 as a citation. We have also added a new reference 90 given that we used information regarding chromosomal gains and losses from the CytoConverter database (https://bmcbioinformatics.biomedcentral.com/articles/10.1186/s12859-019-3062-4).

4. Line 242, gene names are in different formats, for information only to reformat

Thank you for this comment. Apologies but we are unable to identify the specific difference that is being referred to. In relation to point 5 below, we have gone through the manuscript and can confirm that all genes are in uppercase and italicised, however we would be happy for specific changes to be made where identified or for corrections to be made according to journal style at the proofing stage should the article be accepted for publication.

5. In section 4, many gene names need to be Italicized, this is a general rule and need to be followed throughout the manuscript

We thank the reviewer for detecting this oversight. The text has been updated with correct italicization in the following locations:

Page 15, line 274: “IKZF1”

Page 15, line 275: “IKZF1”

Page 15, line 276: “IKZF1 deletion plus other co-occurring deletions such as CDKN2A, PAX5 or PAR1

6. Line 305, one reference is mentioned as recent but is from 2015, better delete recently from the sentence, it is a decade old reference

Thank you for revealing this discrepancy. We have now removed “recently” from this sentence (now on line 295, page 16).

7. Line 307, the year mentioned as 2009 does not match with reference year 2010.....Nature, 2010, 464......

We agree with this point and have updated the year to 2010 accordingly (now on line 298, page 16).

  1. Line 456, Rb1 is mentioned as Rb, need to fix typo.

Thank you for picking up this typo. We have made the correction to “Rb1” (now on line 447, page 24).

Reviewer 2 Report

Comments and Suggestions for Authors

The authors analysed the copy number alteration (CNA) phenomenon in cancer, focusing in particular onto hemopoietic cancers.

The review is very well written and addresses an important and complex theme summarizing the most recent CNA analysis advancements and topics.

The paper is not an original article, but a review. References are appropriate. The tables and figures are as good as the quality of the data.

I have just one minor point to make the article suitable for publication: I would suggest the authors to discuss the Optical Genome Mapping approach in the section “CNA/CNV Detection” as well as in the other ones, since it seems one of the more promising techniques for CNA detection and some interesting results are already published, especially in the haematological context.

Author Response

Comments from Reviewer 2:

The authors analysed the copy number alteration (CNA) phenomenon in cancer, focusing in particular onto hemopoietic cancers.

The review is very well written and addresses an important and complex theme summarizing the most recent CNA analysis advancements and topics.

The paper is not an original article, but a review. References are appropriate. The tables and figures are as good as the quality of the data.

I have just one minor point to make the article suitable for publication: I would suggest the authors to discuss the Optical Genome Mapping approach in the section “CNA/CNV Detection” as well as in the other ones, since it seems one of the more promising techniques for CNA detection and some interesting results are already published, especially in the haematological context.

We thank the reviewer for suggesting this important update. We have now included reference to optical genome mapping in our review. The following text has been added to the revised manuscript (lines 90-96, page 5):

 “Furthermore, the development of optical genomic mapping (OGM) that uses chromosome band patterns from single DNA strands, assembled bioinformatically to identify genomic alterations at high resolution (de novo assembly: 500bp), now offers efficient analysis of genetic alterations within a genome without sequencing (ref 21). Notably, OGM provides an effective tool to detect CNA in cancer, as shown in haematological malignancies (ref 22), and may represent a cost-effective option for genetic screening in future clinical settings (ref 23).”

We have also added OGM into Figure 1 as a potential new assay to detect CNA in cancer.